# Serologic Investigation on Tick-Borne Encephalitis Virus, Kemerovo Virus and Tribeč Virus Infections in Wild Birds

**DOI:** 10.3390/microorganisms10122397

**Published:** 2022-12-02

**Authors:** Katarína Peňazziová, Ľuboš Korytár, Ivana Cingeľová Maruščáková, Petra Schusterová, Alexander Loziak, Soňa Pivka, Anna Ondrejková, Juraj Pistl, Tomáš Csank

**Affiliations:** 1Department of Microbiology and Immunology, University of Veterinary Medicine and Pharmacy in Košice, Komenského 73, 041 81 Košice, Slovakia; 2Department of Epizootiology, Parasitology and Protection of One Health, University of Veterinary Medicine and Pharmacy in Košice, Komenského 73, 041 81 Košice, Slovakia; 3Institute of Social Sciences of the Centre of Social and Psychological Sciences Slovak Academy of Sciences, Karpatská 5, 040 01 Košice, Slovakia

**Keywords:** tick-borne encephalitis virus, Kemerovo virus, Tribeč virus, flavivirus, tick-borne orbivirus, zoonotic, wild birds, ticks, neutralization antibodies

## Abstract

The present study reports on serosurvey on the tick-borne encephalitis virus European subtype (TBEV; genus *Flavivirus*), and the tick-borne Kemerovo (KEMV) and Tribeč (TRBV) orbivirus (genus *Orbivirus*) infections in tick-infested and non-infested birds. No virus RNA was detected in the blood clots. Birds were infested mostly by *Ixodes ricinus*, but *Haemaphysalis concinna* and *I. frontalis* were observed too. TBEV, KEMV and TRBV neutralising antibodies (NAb) were detected in the screening microtitration neutralisation test (μVNT). Seropositive samples were further examined in simultaneous μVNT to distinguish TBEV infection from WNV and USUV. KEMV and TRBV infections were also further examined by μVNT against each other. The demonstrated results point to increased TBEV and TRBV seroprevalence in birds over the past several years. This is the first study on KEMV infection in the Slovak bird population, and seropositive juvenile birds suggest its occurrence in a new geographic area. The results indicate the significance of tick infestation rates, seropositivity and specific NAb titre. The reservoir role of birds for TBEV, KEMV and TRBV remains unclear. However, targeted monitoring of birds and vectors is an effective measure of surveillance of arbovirus introduction into new geographic areas.

## 1. Introduction

The autochthonous Central European bird species belong to the Palaearctic-Afrotropical migrants. Birds are frequently infested with ticks and can transmit them and tick-borne pathogens along the migratory routes [1,2].

Arboviruses (arthropod-borne viruses) are a group of viruses biologically transmitted by blood-sucking arthropods. To date, three species of tick-borne arboviruses have been identified in Slovakia: *Tick-borne encephalitis virus* (TBEV; genus *Flavivirus*; family *Flaviviridae*) [3], *Great Island virus* (GIV; genus *Orbivirus*; family *Sedoreoviridae*) [4] and *Uukuniemi uukuvirus* (UUKV; genus *Uukuvirus*; family *Phenuiviridae*) [5]. Birds are considered potential reservoirs of TBEV, GIV and UUKV [6,7,8,9].

TBEV is the causative agent of tick-borne encephalitis (TBE) and is considered the most important representative of arboviruses in Eurasia, of which the main vector is *Ixodes ricinus* [10]. Although rodents are its main reservoirs, the participation of birds in the natural transmission cycle of TBEV is becoming increasingly important. The magnitude of TBEV viremia in birds depends on the infected species [11]. No viremia was described in great tits (*Parus major*), common pheasants (*Phasianus colchicus*), common kestrels (*Falco tinnunculus*) and common buzzards (*Buteo buteo*); mild viremia was observed in house sparrows (*Passer domesticus*), common quails (*Coturnix coturnix*), mallards (*Anas platyrhynchos*) and common redpolls (*Acanthis flammea*) [12,13,14]. TBEV RNA has been detected in the brain of a buzzard [15]. Sporadically, TBEV was isolated from various bird species, such as redwings (*Turdus iliacus*), western jackdaws (*Coloeus monedula*), carrion crows (*Corvus corone*), Eurasian magpies (*Pica pica*), common starlings (*Sturnus vulgaris*) and other predominantly forest passerines; for more information see Hubálek and Rudolf (2012) [16]. The isolation of TBEV from eggs has been observed in redwing, fieldfare (*Turdus pilaris*), red-throated thrush (*Turdus ruficollis*), pale thrush (*Turdus pallidus*), brown shrikes (*Lanius cristatus*), chestnut-eared buntings (*Emberiza fucata*), Eurasian wrens (*Troglodytes troglodytes*) and northern goshawks (*Accipiter gentilis*) [17]. However, reservoir potential for TBEV was demonstrated only in fieldfares, bramblings (*Fringilla montifringilla*) and common redstarts (*Phoenicurus phoenicurus*), where a high prevalence (50%) of the TBEV RNA and antigen was confirmed [6].

The contribution of birds in the transmission of zoonotic GIV serotypes remains unknown. To date, two GIV serotypes have been reported in Slovakia—Tribeč virus (TRBV) and Lipovník virus (LIPV). *I. ricinus* is considered the main vector of TRBV and rodents as their reservoirs [18,19]. However, seroconversion has been reported in birds [20]. Kemerovo virus (KEMV) is another zoonotic serotype of GIV [21]. It was isolated from the blood of a common redstart caught in Egypt during migration. The bird was likely infected in Eurasia, where the main vector of KEMV, *Ixodes persulcatus,* is widely distributed [8].

Despite the reports on their neurotropic potential, tick-borne orbiviruses are neglected arboviruses, which do not receive much attention. TRBV-specific antibodies were detected in cases of febrile illness and aseptic meningitis [22]. KEMV was isolated from patients with meningitis and meningoencephalitis after tick bites [21]. Nonspecific symptoms make arboviral differential diagnosis problematic. The importance of tick-borne orbiviruses is also underlined by the fact that in Slovakia, between 2016 and 2018, 40% of viral central nervous system infections were diagnosed as unspecified viral encephalitis, meningitis or unspecified viral CNS infection [23].

Although animal diseases caused by TRBV and KEMV have not been reported yet, their role in the transmission cycle and maintenance in nature is not well understood. The aims of the present study build on previous results of arbovirus infection screening in wild birds captured in the model area Drienovská wetland [15,20]. Here, we examined wild living birds’ blood clots for TBEV, TRBV and KEMV RNA and serum samples for virus-specific NAb. Due to the high rate of serological cross-reaction among flaviviruses co-circulating in Central Europe (TBEV, West Nile virus and Usutu virus) and among orbiviruses of the GIV serogroup, the seropositive samples in the screening were further examined by simultaneous microtitration virus neutralization test (μVNT). The relationship between tick infestations and the prevalence and titre of specific NAb were also investigated.

## 2. Materials and Methods

### 2.1. Description of the Model Area

Drienovská wetland is located in the south-east of Slovakia, near the village Drienovec in Košice-okolie District, in the square DFS 7391, at an altitude of 190 m above sea level (Figure 1). In terms of altitudinal division, the wetland is in the lowland (planar stage). From the north, there is an immediate continuity with the hilly area of the Slovak Karst National Park. The wetland has an area of 7.7 ha. Orographically, it is located directly on the border of the Košice Basin and the Slovak Karst. The geographical coordinates of the site are 48°37′ N, 20°55′ E. Habitats surrounding the capture site within 500 metres are the following: 40% arable land, the most common cultivated crops are cereals and sunflower, a smaller part is made up of mown meadow and ruderal habitats, 20% xerothermic vegetation of the foothills of the plateau with shrub formations of pasture character, 20% oak forest, 15% willows (*Salix* sp.) and acrophytes (*Phragmites australis*) and 5% shrub and tree group habitats outside the forest [24].

The Drienovská wetland is known for its bird species richness and population density of migrating and nesting birds, and it is a core locality for the Drienovec Bird Ringing Station [15,20,25,26].

### 2.2. Capturing of Birds and Sample Collection

The birds were captured and handled by a licenced ornithologist (Ľ.K.) under exemption No. 3320/2019-6.3 from Act. No. 543/2002 of the code on nature and landscape protection, granted by the Ministry of Environment of the Slovak Republic.

Ornithological mist nets (Ecotone, Poland) were exposed in the northern part of the wetland in an area of approximately 2.5 ha. Blood samples were collected from transmigrating birds during spring and autumn ringing campaigns in 2019 and 2020. Samples from local breeders and hatched juveniles were collected during the 2019 and 2020 bird nesting seasons (May–July) using the Constant Effort Site (CES) Ringing method according to the British Trust for Ornithology [27] adapted according to [24]. We collected one blood sample per captured bird from fledged juveniles and adults.

For each captured bird, the species, age and, if possible, the sex were determined. All captured and sampled birds were ringed and weighed.

Blood samples were obtained by puncture of the right jugular vein according to [28]. Blood volume no higher than 0.8% of the body weight was collected from each bird. The puncture and blood collection were carried out using an insulin syringe BD Microfine Insulin 0.5 mL with a U-100 needle (Becton Dickinson & Comp., Franklin Lakes, NJ, USA). Immediately after sampling, birds were released at the capture site.

Ticks were removed from the birds using a tick removal spoon (Dr. Kapiller^®^, Budapest, Hungary) (Appendix A). We introduced the use of this tick removal spoon because it has a round shape and eliminates the risk of losing the tick immediately after removal from the bird.

### 2.3. Tick Diagnostics

Ticks were placed in tubes labelled with the host bird’s number and collection date and promptly transferred to the deep freezer (−80 °C). Morphological identification was performed by an SZO-4 stereomicroscope (Optika, Ponteranica, Italy) according to the morphological key [24,28].

### 2.4. Processing of Blood Samples

Blood samples collected during the first day of trapping according to CES Ringing methodology were stored overnight in the field at refrigerator temperature. Sera were separated by centrifugation using 3800 RCF at 4 °C for 30 min. The sera were collected into new tubes and stored with the blood clots at −80 °C until examination.

Blood clots were processed into a 10% (*w*/*v*) suspension in Eagle’s minimum essential medium (EMEM; Pan Biotech, Aidenbach, Germany). Suspensions were centrifuged at RCF 13,000 RCF at 4 °C for 10 min, and the supernatants were used for viral RNA isolation.

### 2.5. RNA Isolation and RT-PCR for Arbovirus Detection

Only blood clots of tick-infested individuals were tested for TBEV, TRBV and KEMV RNA. Nucleic acid was extracted using the NucleoSpin RNA virus kit (Macherey Nagel, GmbH & Co., Dueren, Germany) according to the manufacturer’s instructions and kept at −80 °C until use.

The complementary DNA was synthesised using LunaScript RT SuperMix (New England Biolabs, Ipswich, MA, USA) according to the manufacturer’s protocol. The obtained cDNA was stored at −20 °C and used as a PCR template for the molecular detection of arboviruses.

Orbivirus RNA was detected by conventional PCR and flavivirus RNA was detected by hemi-nested PCR using the DreamTaqTM Green PCR Master Mix (Thermo Fisher Scientific, Vilnius, Lithuania). To detect TRBV and KEMV RNA, Orbi_GIV_serogr_F and Orbi_GIV_serogr_R primers amplifying a 770 bp PCR product in the sequence VP1 located on segment 1 were used [29]. PanFlavi-NS5-F [15] and cFD2 [30] primers used in the first flavivirus PCR amplify a 599 bp sequence in the NS5 protein. In the hemi-nested flavivirus PCR, the PanFlavi-NS5-F and PanFlavi-NS5-R [15] primers flanking a 360 bp PCR product were used. Each primer was used in a 400 nM final concentration. The thermal profiles of each PCR reaction are described in Appendix A.

### 2.6. μVNT for Arbovirus NAb Screening

Serum samples were first screened by μVNT for TBEV, TRBV and KEMV seropositive individuals. Heat-inactivated samples (56 °C for 30 min) were diluted 1:5 in a volume of 25 μL. Diluted sera were mixed with 25 μL of the virus culture containing 100 TCID_50_, giving a final serum dilution of 1:10. Viruses used in the μVNT were the following: TBEV Hypr (kindly provided by Dr. Mária Takács, National Centre for Epidemiology, Budapest, Hungary), KEMV (kindly provided by Professor Gerhard Dobler, Institut für Mikrobiologie der Bundeswehr, München, Germany) and TRBV strain 16.C/16 [31]. After overnight incubation at 4 °C, 50 μL of the cell suspension containing 1 × 10^4^ cells in Eagle’s Minimal Essential Medium (Biosera, Nuaillé, France) supplemented by 10% foetal bovine serum (Biosera) and antibiotics were added to each well. Vero E6 cells were used for KEMV and TRBV, and A549 cells for TBEV. Plates were incubated at 37 °C in a 5% CO_2_ atmosphere for three (TRBV and KEMV) and five (TBEV) days. Results were read by an inverted light microscope at 100–200× magnification.

### 2.7. Simultaneous μVNT for Differentiation of Arbovirus Infections

Due to the high serological cross-reaction among flaviviruses and GIV serogroup orbiviruses, the seropositive samples from the screening μVNT were further examined by simultaneous μVNT to differentiate the infection. Here, serum samples were 4-fold serially diluted to 1:10,240. The abovementioned orbiviruses were used for KEMV and TRBV infection differentiation. For the differentiation of flavivirus infections, TBEV strain Hypr, Usutu virus (USUV) strain 939/01 (kindly provided by Professor Norbert Nowotny, Veterinary University, Vienna, Austria) and West Nile virus (WNV) strain 291.B/2013/Velky Biel/SVK [32] were used. In cross-reactive serum samples, at least a 4-fold higher specific NAb titre was considered conclusive.

During each sample batch in the screening and simultaneous μVNTs, the virus inoculum was back-titrated in triplicates, and the average titre was considered the infective dose.

### 2.8. Quantitative Characteristics of the Captured Bird Population

Using the ecological index of dominance (IED%), we characterised the population of captured birds [33,34]:$$\mathrm{IED}=\frac{\mathrm{NPB}}{\mathrm{NB}}\times 100\%$$

IED—ecological index of bird species dominance, NPB—number of birds of a particular species, NB—total number of birds.

Birds were infested when at least one tick was attached. The prevalence of tick infestation in a bird species (PTI%) and the mean intensity of tick infestation in a certain species (MITB) were calculated according to previous studies [33,34,35]:$$\mathrm{PTI}=\frac{\mathrm{NPBT}}{\mathrm{NPB}}\times 100\%$$

PTI—prevalence of tick infestation per bird species, NPBT—number of birds of a particular species infested with ticks, NPB—number of birds of a particular species.
$$\mathrm{MITB}=\frac{\mathrm{TNPB}}{\mathrm{NPBT}}$$

MITB—mean intensity of tick infestation per bird, TNPB—number of all tick species collected from a particular bird species, NPBT—number of birds of a particular species infested with ticks.

### 2.9. Statistical Analysis

R software (libraries: stats, rstatix, tidyverse) version 4.2.2 was utilised for statistical analysis. Because of the confirmed non-normal distribution of dependent variables, nonparametric tests were used—specifically, Fisher’s exact test (when testing the infestation of birds) and the Kruskal–Wallis H test (when testing the prevalence of ticks). R script (File S1) documenting all statistical analysis and the data used (File S2) are available on request.

## 3. Results

### 3.1. Ornithological and Parasitological Findings

During 15 bird-trapping visits carried out in 2019 and 2020, a total of 393 birds belonging to 32 species were captured. Based on IED data, Eurasian blackcaps (*Sylvia atricapilla*) and European robins (*Erithacus rubecula*) dominated the bird species list (23.7%), followed by the great tit (18.3%; *n* = 72) and common blackbird (*Turdus merula*) (7.9%; *n* = 31). A detailed list of caught bird species is in Appendix A.

Tick infestation was observed in 22.4% (*n* = 88) of caught birds belonging to nine species (Table 1 and Appendix A). The most frequently infested species, according to PTI, was the common blackbird (87.1%; 27/31), followed by song thrush (53.8%; 7/13), dunnock (*Prunella modularis*) (40%; 2/5), common chaffinch (*Fringilla coelebs*) (27.3%; 3/11), great tit (25%; 18/72), European robin (24.7%; 23/93), hawfinch (*Coccothraustes coccothraustes*) (21.4%; 3/14) and the common nightingale (*Luscinia megarhynchos*) (20%; 18/72) (Table 1).

In total, 194 ticks were collected from birds. *Ixodes ricinus* was found in 95.9% (*n* = 186) of examined cases, followed by *Haemaphysalis concinna* in 1% (*n* = 2) and *I. frontalis* in 0.5% (*n* = 1). Due to the damage of ticks, 2.6% (*n* = 5) were recognised at the genus level as *Ixodes* spp. Nymphs were found in 71.1% (*n* = 138; *I. ricinus*, *n* = 134; *I. frontalis*, *n* = 1; *Ixodes* spp., *n* = 3), and the rest (28.9%; *n* = 56) were larvae (*I. ricinus*, *n* = 52; *H. concinna*, *n* = 2; *Ixodes* spp., *n* = 2) (Table 1).

The highest MITB was observed in blackbirds (3.44 ticks per bird), followed by hawfinches (2.3 ticks per bird), European robins (2.1 ticks per bird), common nightingales and dunnocks with equal MITB (2.0 ticks per bird) (Table 1).

We compared whether migratory status, species, sex, age, or feeding behaviour could influence tick prevalence using the Kruskal–Wallis H tests (Table 2). Data from nine species (Table 1 and File S2 avalaible on request) where at least one bird was infested with ticks were used for statistical analysis.

Migratory status was a significant factor determining tick prevalence, whether it was the total number of ticks (*p* ≤ 0.0000) or larvae (*p* ≤ 0.0177) or nymphs (*p* ≤ 0.0000). The highest number of ticks was observed for short-distance migrants (S), whereas the lowest number was observed in bird species in which some individuals are short-distance migrants and others are long-distance migrants (S/L). Pairwise comparisons also confirmed a significant difference between these groups (*p* ≤ 0.0000—total ticks and nymph prevalence; *p* ≤ 0.0082—larvae prevalence) (Table 2).

Another relevant factor determining the prevalence of ticks, larvae and nymphs was the feeding behaviour of the birds. We divided all infested species into two groups: Ground-feeding species (common blackbird, song thrush) and shrub-feeding species (hawfinch, European robin, common chaffinch, common nightingale, great tit, dunnock, common blackcap). Ground-feeding birds had a significantly higher prevalence of ticks (*p* ≤ 0.0000), larvae (*p* ≤ 0.0000) and nymphs (*p* ≤ 0.0000) than those feeding in shrubs (Table 2).

When comparing infestation in species, only hawfinches and common blackbirds showed a statistically significant difference in tick prevalence (*p* ≤ 0.0000) (Table 2).

The ages and sexes of birds were evaluated as factors not influencing tick prevalence (Table 2).

Fisher’s exact tests were used to determine the differences between tick-infested and non-infested birds within the selected nine species (the same species used for the Kruskal–Wallis H test). We investigated whether the infestation is influenced by migratory status, feeding behaviour, sex and age. A significant difference (*p* ≤ 0.0000) between groups was confirmed for migration status, where S migrants were more likely infested than S/L migrants, and feeding behaviour (*p* ≤ 0.0000), where ground-feeding birds were more likely infested than shrub-feeding birds. The effect of sex and age on infestation was not confirmed (Table 3).

### 3.2. Screening of Arbovirus Infections

None of the 88 blood clot samples collected from tick-infested birds were positive for the screened arbovirus RNA.

In total, 393 serum samples were included in the study, 305 from non-infested and 88 from tick-infested birds. However, due to haemolysis, three samples of non-infested and 14 samples of tick-infested bird sera were excluded from the neutralization assay. Overall, in the μVNT, 302 non-infested and 74 tick-infested serum samples were used in the screening for TBEV, KEMV and TRBV NAb at a serum dilution of 1:10 (Appendix A). Due to the limited volume, not all the serum samples were screened for each arbovirus NAb.

TBEV NAb was detected in 9.8% (*n* = 37) of 376 tested sera, collected from 302 non-infested and 74 tick-infested birds (Appendix A). Seropositivity was observed in 10.6% (*n* = 32) of the non-infested (common blackbird, *n* = 2; common nightingale, *n* = 1; Eurasian blackcap, *n* = 24; European robin, *n* = 5) and in 6.8% (*n* = 5) of tick-infested (common blackbird, *n* = 4; European robin, *n* = 1) individuals. Juveniles represented 40.5% (*n* = 15) of all TBEV NAb seropositive birds (common blackbird, *n* = 3; Eurasian blackcap, *n* = 11; European robin, *n* = 1). 

KEMV NAb seroprevalence was detected in 7.5% (*n* = 24) of 318 tested sera collected from 244 non-infested and 74 tick-infested birds (Appendix A). Seropositivity was observed in 6.6% (*n* = 16) of non-infested (common chaffinch, *n* = 2; Eurasian blackcap, *n* = 7; Eurasian jay, *n* = 2; European robin, *n* = 2; great tit, *n* = 2; marsh warbler, *n* = 1) and in 10.8% (*n* = 8) of tick-infested (common blackbird, *n* = 5; European robin, *n* = 1; great tit, *n* = 1; hawfinch, *n* = 1) individuals. Juveniles represented 33.3% (*n* = 8) of all KEMV NAb seropositive birds (common blackbird, *n* = 2; Eurasian blackcap, *n* = 3; European robin, *n* = 2; great tit, *n* = 1).

TRBV NAb seroprevalence was observed in 19.5% (*n* = 50) of 256 tested sera collected from 182 non-infested and 74 tick-infested birds (Appendix A). Seropositivity was observed in 14.3% (*n* = 26) of the non-infested (common blackbird, *n* = 1; common chaffinch, *n* = 1; Eurasian blackcap, *n* = 4; Eurasian jay (*Garrulus glandarius*), *n* = 1; European robin, *n* = 6; great tit, *n* = 8; hawfinch, *n* = 2; marsh warbler (*Acrocephalus palustris*), *n* = 2; redwing, *n* = 1) birds. In the tick-infested group, TRBV NAb was detected in 32.4% (*n* = 24; common blackbird, *n* = 10; common chaffinch, *n* = 1; dunnock, *n* = 1; European robin, *n* = 4; great tit, *n* = 1; hawfinch, *n* = 3; song thrush, *n* = 4) individuals. Juveniles represented 42% (*n* = 21) of all TRBV NAb seropositive birds (common blackbird, *n* = 6; Eurasian blackcap, *n* = 1; European robin, *n* = 5; great tit, *n* = 7; song thrush, *n* = 2). 

### 3.3. Determination of Arbovirus Infections by Simultaneous μVNT

Sufficient volume for simultaneous μVNT was available in 9 sera out of the 37 TBEV NAb positive samples (Table 4). A low TBEV NAb titre (1:10) was observed in five serum samples collected from non-infested birds. The remaining four samples were collected from tick-infested blackbirds, where the TBEV NAb titre reached 1:10–1:40. Two of these samples cross-reacted with WNV (86.B/19 and 118.B/19) and one with USUV (86.B/19). Sample 86.B/19 showed a 4-fold higher NAb titre (1:160) in favour of USUV, which might indicate USUV infection. In the case of serum 118.B/19, the differentiation of TBEV and WNV infections was not possible due to the equal titre of NAb (Table 4).

The determination of orbivirus infection was possible in 27 serum samples. Eight samples were collected from non-infested and 19 from infested birds (Table 5).

One non-infested Eurasian jay (49.B/20) tested positive only for KEMV with the 1:10 NAb titre (Table 5). Twenty-four birds tested positive for TRBV infection with NAb titres ranging from 1:10 to 1:10,240. In seven samples, the TRBV NAb titre ranged from 1:10 to 1:40, in ten birds from 1:40 to 1:640, and in seven birds, the titre was higher than 1:640. A cross-reaction with KEMV was observed in six samples, but in each case, the TRBV NAb titre was at least 4-fold higher.

Determination was not possible in two birds, a hawfinch (5.B/20) and a European robin (279.B/20). The hawfinch had equal TRBV and KEMV NAb titre (1:10), and in the latter sample, the endpoint titration of KEMV NAb was not possible due to the low volume of serum.

### 3.4. Simultaneous Occurrence of Flavivirus and Orbivirus NAb

Two serum samples collected from adult common blackbirds (86.B/19 and 216.B/19) contained NAb against flaviviruses and TRBV (Table 4 and Table 5). Namely, sample 86.B/19 was simultaneously positive for USUV NAb (1:160) and TRBV NAb (1:2560). Sample 216.B/19 tested positive for low TBEV NAb titre (1:10–1:40) and high titre of TRBV NAb (1:2560). 

### 3.5. Statistical Evaluation of the Link between Tick Infestation, Seropositivity and Arbovirus NAb Titre

A difference between groups of tick-infested and non-infested birds in seropositivity (results from screening, Appendix A) was statistically significant only in TRBV infection (*p* ≤ 0.0016, Table 6). Statistical results indicate that tick-infested birds were more likely to overcome TRBV infection than non-infested birds. Statistical significance between the NAb titre and tick infestation was observed in the case of the TBEV (1:10 or higher) titre (*p* ≤ 0.0119) and TRBV 1:160 or higher titre (*p* ≤ 0.0261), where tick-infested birds have a higher NAb titre than non-infested individuals (Table 6).

## 4. Discussion

The present study reports on the participation of birds in the transmission of TBEV, KEMV and TRBV infections and the link between tick infestation and arbovirus infections under natural conditions. 

During 2019 and 2020, 393 birds were captured, of which 22.4% were infested with ticks. The most abundant (69.1%) ticks were *I. ricinus* nymphs. Similar to the abovementioned publications, in our study, the most infested were ground-feeding or medium-level foraging bird species (common blackbird, song thrush, dunnock, common chaffinch, great tit, European robin, hawfinch and the common nightingale). These results are comparable to previous studies from Slovakia [36,37], where nymphs feeding on birds were more common than larvae. However, in another study from Slovakia, the trend was in favour of larvae [38]. Except for *I. ricinus*, we also collected two tick species, *H. concinna* and *I. frontalis*, rarely obtained from birds in Slovakia [36,37,38].

Studies focusing on tick-borne arbovirus infections were carried out in autochthonous wild bird populations in Slovakia. Approximately 400 migratory, partially migratory and non-migratory birds of more than 20 species were tested for antibodies [15,20,25]. TBEV NAb was observed in an adult Eurasian blackcap, representing 1.1% seropositivity [20]. In comparison with our previous study [20], the present results show that TBEV NAb prevalence increased by almost 26% among the Eurasian blackcaps. Some individuals of the Eurasian blackcap population are short-distance migrants and others are long-distance migrants [39]. Thus, it is not possible to confirm that the birds seroconverted at the model area in the Drienovská wetland. Their nests are neat cups built low in brambles or scrubs close to the ground [40]. Indeed, adults and juveniles have a high probability of coming into contact with infected ticks.

Eleven TBEV NAb-positive Eurasian blackcaps were juveniles. The low TBEV NAb titre in the examined birds may be related to either developing the post-infectious antibody response or antibody decay of passively acquired maternal antibodies. Knowledge of post-infection antibody development and decay after flavivirus infections is limited. The nature and duration of the antibody response may vary between populations and bird species [41,42,43,44]. Haemagglutination-inhibiting antibodies were detected in Western capercaillie (*Tetrao urogallus*), willow ptarmigan (*Lagopus lagopus*) and rock ptarmigan (*Lagopus muta*) after louping ill virus (LIV) infection, a close relative virus to TBEV. LIV antibodies were detected (1:40–1:160) six days after infection, and in four days, reached high titre levels ranging from 1:640 to 1:10,240 [45]. In free-ranging birds, WNV NAb decreased by 0.188 log natural units per month. Most birds had an undetectable NAb titre two years following initial exposure to WNV, and juveniles had higher antibody decay rates than adults [46].

Other TBEV Nab-positive bird species included the European robin, common blackbird and a common nightingale. Several studies have described European robins and common blackbirds as potential TBEV reservoirs. Blackbirds in particular were highly infested in the present study. The presumption that these species could be reservoirs for TBEV stemmed mainly from the finding of virus-positive ticks on birds and TBEV NAb in blood samples [47,48,49,50]. However, isolation or detection of TBEV from bird blood is usually unsuccessful. In the present study, we also failed to detect virus RNA in the feeding ticks and in tick-infested bird blood clots. This result concurs with previous studies, where the prevalence of TBEV in endemic areas in questing ticks and ticks removed from hosts is less than 1% [51,52,53,54].

Both KEMV and TRBV are considered serotypes of GIV and are serologically close relatives. Differentiation of specific antibodies is possible based on the complement fixation test or neutralisation assay [55,56]. The main reservoirs of GIV are seabirds and small mammals of KEMV and TRBV [18]. The role of birds in the transmission of these viruses remains unclear. To date, there is only one report on the successful isolation of KEMV, the EgAn1169-61 strain, from the blood of a migrating redstart [8]. KEMV NAb was demonstrated in 37% sera of juvenile and adult wild birds caught near Romanovka village (Kemerovo region, Russia), where the original KEMV strains were recovered from *I. persulcatus* ticks. KEMV NAb specific to the R10 strain was detected in fieldfare, red-throated thrush, song thrush, carrion crow, common starling, tree pipit (*Anthus trivialis*), common buzzard and Eurasian magpie. The titre of NAb ranged from 1:6 to 1:8, and the highest seropositivity rate was noted in fieldfares [57]. Despite the high number of examined samples in our study, not all could be used for simultaneous μVNT due to the limited volume of sera, and KEMV infection was conclusively differentiated from TRBV infection only in one adult Eurasian jay. In one adult hawfinch, differentiation was not possible because of equal NAb titres.

The high seroprevalence in the study of Libíková et al. was likely caused by the origin of tested birds, which were caught in the area of the natural occurrence of KEMV [57]. KEMV NAb-seropositive adult Eurasian jay and dubious hawfinch caught in the Drienovská wetland may indicate that these birds migrated from endemic areas where they seroconverted earlier, or they become infected in Central Europe. However, reports of hawfinches ringed on the Drienovská wetland support the possibility of KEMV presence in Central Europe [58]. These birds migrate in the southwestern direction, which suggests the presence of KEMV along their migratory routes [59].

Until recently, it was believed that the distribution of KEMV is closely associated with Western Siberia. However, the latest molecular research indicates that KEMV circulates in areas thousands of miles apart (the Urals and certain areas in the European part of the Russian Federation) [60,61,62,63]. Multiple reassortments were observed among nine KEMV strains isolated in distinct parts of Russia. These results point to virus trafficking over long distances and support the assumption that birds are crucial in spreading the virus to new areas [62].

In the screening, nine sera of non-infested birds (marsh warbler, common chaffinch and Eurasian blackcaps) and one serum of infested great tit were positive only for KEMV NAb. Three of these birds were juvenile blackcaps. This may suggest previous KEMV infection of juveniles in Northeastern or Eastern Europe, where the virus is endemic. The birds may become seropositive by acquiring antibodies through eggs or infection in Central Europe. If the birds became infected in central Europe, that would suggest two hypotheses. First, the natural vector of KEMV *I. persulcatus* has already spread into a new area. According to the Tick maps of the European Centre for Disease Prevention and Control, there are no data on the presence of this tick species in Slovakia, Poland or Hungary. However, *I. persulcatus* was observed in the north of Ukraine [64]. Second, KEMV may have an alternative vector by which it is able to multiply and be transmitted to birds. A recent study showed, that an artificial feeding system allowed *I. ricinus* to acquire KEMV and transmit it transstadially, but the ticks could not transmit the virus to IFNAR^-/-^ or BALB/c mice [65]. However, the transmission of KEMV by *I. ricinus* to birds is unknown.

The highest seropositivity among the screened arbovirus infections was observed in the case of TRBV. Most of the NAb-quantified samples had titres equal to or higher than 1:160, and some juvenile individuals fell into this group. Compared to previous research by Csank et al., where the prevalence of TRBV NAb was 7.4%, in the present research, it was 19.5% [20]. High NAb titres in adults and juveniles suggest an active transmission cycle of TRBV in the avifauna of the selected area, but also possible infection in other localities.

Circulation of TRBV and LIPV in the locality of Slovak karst has been reported before. In 1963, seven strains of LIPV were isolated from questing and half-engorged *I. ricinus* ticks [19]. Another three strains were isolated from half-engorged and engorged *I. ricinus* ticks [66]. In addition, in Western Slovakia, three further strains named Koliba were isolated from *I. ricinus* ticks [19]. Hence, in the case of samples with equal KEMV and TRBV NAb titres, it is safe to hypothesise that those sera may contain LIPV or Koliba NAb. Unfortunately, these strains are not kept at our disposal, and it is not possible to further examine the bird serum samples.

In two serum samples from adult blackbirds, simultaneous μVNT showed co-infection by flaviviruses and TRBV. Co-infection of flaviviruses and members of the former KEMV serogroup was described in the 1960s. TBEV and KEMV were detected in the cerebrospinal fluid of a patient diagnosed with TBE [21]. Co-infection was also observed in ticks, and LIPV NAb were confirmed in 51% of 49 patients diagnosed with TBE [67]. Recently, we have demonstrated co-infection of WNV and TRBV at the model area in the Drienovská wetland [20]. The benefits of these viruses from co-infection remain unknown.

## 5. Conclusions

The demonstrated results indicate increased TBEV and TRBV seroprevalence in birds over the past several years. This is the first study investigating KEMV infection in the Slovak bird population. The high KEMV seropositivity in juveniles may indicate that the infection occurs in the model locality Drienovská wetland. However, at this stage, it is impossible to distinguish post-infectious seroconversion from passively acquired maternal antibodies. Hence, further research should be aimed at this field. Statistical analysis confirmed the significance of tick infestation rates in TRBV seroconversion. The relationship was demonstrated for both seropositivity and the NAb titre. In the case of TBEV, the infestation rate was associated with the amount of NAb. Although the role of birds as reservoirs of TBEV, KEMV and TRBV remains unclear, targeted monitoring of birds and vectors is an effective measure of the surveillance of zoonotic arbovirus introduction into new geographic areas.

## Figures and Tables

**Figure 1 microorganisms-10-02397-f001:**
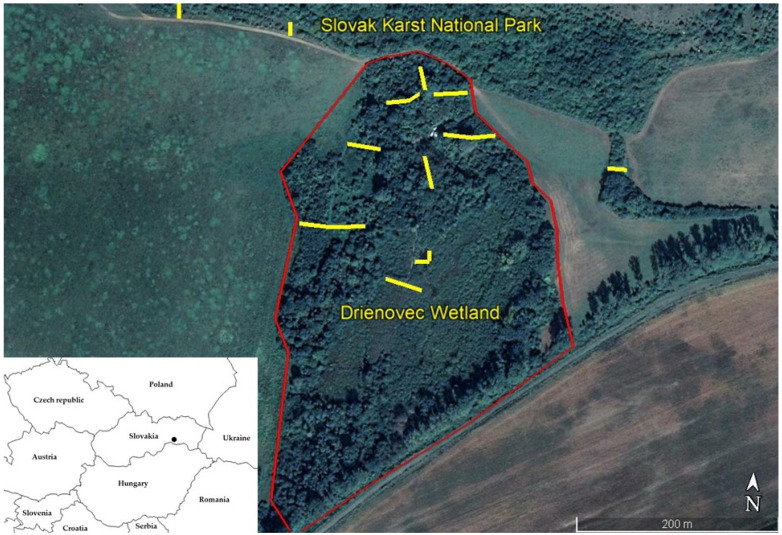
Location of Drienovská wetland and the distribution of mist nets. Legend Yellow lines depict the distribution of mist nets on the model area Drienovská wetland. Red lines depict the border of the model area. The insert in the left lower corner shows the location of Slovakia and its neighbouring countries in Central Europe.

**Table 1 microorganisms-10-02397-t001:** Tick infestation among the analysed bird species.

Bird Species	No. of Caught Birds	IED (%)	MS	No. of Tick Infested Birds	No. of Collected Ticks	Tick Species								PTI (%)	MITB
						*I. ricinus*		*I. frontalis*		*H. concinna*		*Ixodes* spp.			
						La	N	La	N	La	N	La	N		
Hawfinch	14	3.6	S	3	7	5	2	-	-	-	-	-	-	21.4	2.3
European robin	93	23.7	S	23	48	23	23	-	1	1	-	-	-	24.7	2.1
Common chaffinch	11	2.8	S	3	5	1	4	-	-	-	-	-	-	27.3	1.7
Common nightingale	10	2.5	L	2	4	-	4	-	-	-	-	-	-	20.0	2.0
Great tit	72	18.3	S	18	18	1	14	-	-	-	-	-	3	25.0	1.0
Dunnock	5	1.3	S	2	4	1	3	-	-	-	-	-	-	40.0	2.0
Eurasian blackcap	93	23.7	S/L	3	4	3	1	-	-	-	-	-	-	3.2	1.3
Common blackbird	31	7.9	S	27	93	16	74	-	-	1	-	2	-	87.1	3.4
Song thrush	13	3.3	S	7	11	2	9	-	-	-	-	-	-	53.8	1.6

Legend: no.—number; -—not found; IED—index of ecological dominance of bird species; MS—migratory status; L—species strictly long-distance migrant; S—species strictly short-distance migrant; S/L—species in which some individuals are short-distance migrants whereas others are long-distance migrants; I.—Ixodes; H.—Haemaphysalis; spp.—species; La—larvae; N—nymphs; PTI—prevalence of tick infestation per bird species; MITB—mean intensity of tick infestation per bird.

**Table 2 microorganisms-10-02397-t002:** Results of Kruskal–Wallis H tests for tick prevalence in selected bird species where at least one bird was infested with ticks.

Dependent Var.	Factors	*p*-Values	Effect Size (eta^2^)	Group Means (No. of Ticks)				Significant Pairwise Comparisons
Prevalence of ticks	Migration status	*p* ≤ 0.0000 *	0.0962 moderate	S	S/L		L	*p* ≤ 0.0000 *; S vs. S/L
				0.8	0.04		0.4	
Prevalence of larvae	Migration status	*p* ≤ 0.0177 *	0.0179 small	S	S/L		L	*p* ≤ 0.0082 *; S vs. S/L
				0.22	0.03		0	
Prevalence of nymphs	Migration status	*p* ≤ 0.0000 *	0.0713 moderate	S	S/L		L	*p* ≤ 0.0000 *; S vs. S/L
				0.56	0.01		0.4	
Prevalence of ticks	Species (S1–S9)	*p* ≤ 0.0000 *	0.295 large	S1		S8		*p* ≤ 0.0000 *; S1 vs. S8
				0.5		3		
Prevalence of ticks	Sex	*p* ≤ 0.226	-	-				-
Prevalence of ticks	Age	*p* ≤ 0.306	-	-				-
Prevalence of ticks	Feeding behaviour	*p* ≤ 0.0000 *	0.236 large	Ground		Shrubs		-
				2.36		0.3		
Prevalence of larvae	Feeding behaviour	*p* ≤ 0.0000 *	0.049 small	Ground		Shrubs		-
				0.48		0.18		
Prevalence of nymphs	Feeding behaviour	*p* ≤ 0.0000 *	0.210 large	Ground		Shrubs		-
				1.89		0.19		

Legend: no.—number; var.—variables; -—not evaluated; *—significance mark; L—species strictly long-distance migrant; S—species strictly short-distance migrant; S/L—species in which some individuals are short-distance migrants whereas others are long-distance migrants; Species: 1—Hawfinch; 2—European robin; 3—Common chaffinch; 4—Common nightingale; 5—Great tit; 6—Dunnock; 7—Common blackcap; 8—Common blackbird; 9—Song thrush.

**Table 3 microorganisms-10-02397-t003:** Results of Fisher’s exact tests for determining the difference between tick-infested and non-infested birds in selected bird species where at least one bird was infested with ticks in sex, age, feeding behaviour and migration status.

Dependent Variables	Independent Variables (Factors)	*p*-Values
Tick infestation	Sex	*p* ≤ 0.4725
Tick infestation	Age	*p* ≤ 0.3834
Tick infestation	Feeding behaviour	*p* ≤ 0.0000 *
Tick infestation	Migration status	*p* ≤ 0.0000 *

Legend: *—significance mark.

**Table 4 microorganisms-10-02397-t004:** Determination of flavivirus infections by simultaneous μVNT.

Species	Sample	Infestation	Estimated Age	TBEV NAb	WNV NAb	USUV NAb
Common blackbird	86.B/19	+	+1K	1:10–1:40	1:10–1:40	1:160
Common blackbird	118.B/19	+	+1K	1:10–1:40	1:10–1:40	-
Common blackbird	216.B/19	+	+1K	1:10–1:40	-	-
Common blackbird	245.B/20	+	1K	1:10–1:40	-	-
Eurasian blackcap	278.B/19	-	+1K	1:10	-	-
Common blackbird	509.B/19	-	1K	1:10	-	-
Eurasian blackcap	223.B/20	-	+1K	1:10	-	-
European robin	225.B/20	-	+1K	1:10	-	-

Legend: Infestation: +—infested, -—non-infested; Estimated age: 1K—juvenile, +1K—adult.

**Table 5 microorganisms-10-02397-t005:** Determination of TRBV and KEMV infections by simultaneous μVNT.

Species	Sample	Infestation	Estimated Age	TRBV NAb	KEMV NAb
Common chaffinch	60.B/19	+	+1K	1:640	-
Common blackbird	86.B/19	+	+1K	1:2560	-
Common blackbird	216.B/19	+	+1K	1:2560	1:40
Song thrush	224.B/19	+	+1K	1:640	-
Hawfinch	226.B/19	+	+1K	1:2560	-
Hawfinch	230.B/19	+	+1K	1:160–1:640	-
Common blackbird	236.B/19	+	+1K	1:2560–1:10,240	-
Common blackbird	242.B/19	+	+1K	1:10,240	1:10–1:40
Dunnock	266.B/19	+	+1K	1:640	-
Common blackbird	286.B/19	+	+1K	1:640	1:40
Song thrush	513.B/19	+	+1K	1:10–1:40	-
Common blackbird	517.B/19	+	1K	1:10	-
Common blackbird	541.B/19	+	1K	1:640	1:10
Common blackbird	549.B/19	+	1K	1:640	1:10–1:40
Common blackbird	579.B/19	+	1K	1:160	-
Hawfinch	5.B/20	+	+1K	1:10	1:10
Common blackbird	107.B/20	+	1K	1:10	-
Song thrush	215.B/20	+	1K	1:2560–1:10,240	-
Song thrush	235.B/20	+	1K	1:10	-
Redwing	258.B/19	-	+1K	1:10	-
Eurasian jay	425.B/19	-	+1K	1:640–1:2560	1:10
Hawfinch	507.B/19	-	+1K	1:40	-
Common blackbird	529.B/19	-	1K	1:40–1:160	-
Hawfinch	29.B/20	-	+1K	1:10–1:40	-
Eurasian jay	49.B/20	-	+1K	-	1:10
European robin	275.B/20	-	1K	1:10	-
European robin	279.B/20	-	+1K	1:40–1:160	1:10 *

Legend: Infestation: +—infested, -—non-infested; Estimated age: 1K—juvenile, +1K—adult; *—screening result, due to the low amount of serum, endpoint NAb titration was not possible.

**Table 6 microorganisms-10-02397-t006:** Results of Fisher’s exact tests for determination of the difference between tick-infested and non-infested birds in seroprevalence and antibody titre.

Dependent Variables	Independent Variables (Factors)	*p*-Values
Tick infestation	Seroprevalence TRBV	*p* ≤ 0.0016 *
Tick infestation	Seroprevalence KEMV	*p* ≤ 0.2186
Tick infestation	Seroprevalence TBEV	*p* ≤ 0.3895
Tick infestation	TRBV antibody titre ≥1:160	*p* ≤ 0.0261 *
Tick infestation	KEMV antibody titre ≥1:10	*p* ≤ 0.4286
Tick infestation	TBEV antibody titre ≥1:10	*p* ≤ 0.0119 *

Legend: *—significance mark.

## Data Availability

File S1 R script and File S2 Data used for statistical analysis are available on request.

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
