# Peer review of "Serologic Investigation on Tick-Borne Encephalitis Virus, Kemerovo Virus and Tribeč Virus Infections in Wild Birds"

_microorganisms, 2022, doi:10.3390/microorganisms10122397_

Round 1

Reviewer 1 Report

Due to the increasing incidence of tick-borne diseases, surveillance of tick-borne diseases is an important factor to consider in terms of their impact on human and animal health.

The authors have carried out their study on the differents wild bird species that may carry these diseases in central-eastern Europe. They focused on three tick-borne arbobiruses encephalitis , Kemerovo and Tribeč virus.

These works are necessary in the framework of  Onehealth, and complex, especially when they are performed on wild animals.

however some changes and improvements should be made before the publication of the work:

Line 90 described briefly the aerea

Line 96 described briefly the protocol for sampling blood

Lines 120-128 descrieb PCR protocol with the aniling temperatures

Line 211 – 218 Taking in account the complicity and the difficulties with the biological samples, authors should be include more than 9 samples to give a robustness resutls.

Authors should include serological antibody studies using an indirect enzyme-linked immunosorbent assay against arbovirus proteins. This will give an idea of how many wild birds were at some time exposed to these viruses. It would give a first approximate result on the epidemiology of these arboviruses in wild birds in their research region.

Reviewer 2 Report

Serologic investigation on tick-borne encephalitis virus, Kemerovo virus and Tribeč virus infections in wild birds

The paper concerns serious problem with viruses occurrence in various bird species in Slovakia. Paper is very interesting, but I have few commments.

Main comments:

You must provide latin name for every bird species mentioned, not only common name.

In many lines in the MS authors didnt cited source of date (a lot of references are missing) ie: line 40, 42, 43, 56, 66, 71 and many places in discussion section

Also, I am wondering if some newer references are available. In the bibliography section, 17 of 62 references are very old. Also, some data from ref list originate from Slovak conference and is therefore hard to find. I highly recommend refreshing the reference list, as far as it is possible.

Materials and methods:

The study area is poorly described – it would be a good idea to include a map of the whole survey area. It would make the paper much more reader- friendly. Also, the methods of birds trapping should be at least briefly described- it is difficult to find all necessary informations in other papers- general inrofmation should be provided in the text, especially when birds were examined.

Microtitration virus neutralisation test- the description of methods in this paragraph is very complicated and difficult to follow

It would be nice to add a photo of a tick removal spoon and why was this tool used? Is it recommended for collecting ticks from birds?

Line 104-106 I do not understand this method of sera collection, why was the blood incubated overnight?

It is not clear whether you isolated DNA from collected ticks or not? Have you also tested DNA/RNA material obtained from collected ticks?

I have serious doubts about statistical methods, especially the part regarding tick analysis. Which part of the data was calculated in MS Excel? For statistical analysis please use such programs as R, STATISTICA, SPSS statistic. Excel and Graphrism in this case can be insufficient tools.

Result section

This section must be improved. The main complaint concerns lack of statistical values. Statistical data is missing almost throughout the paragraph. The authors calculated some tick prevalence (without statistical data) but not mean abundance of tick infestation (statistical analyzes should include tick species, tick stadium, bird stadium, bird age and, what can be the most interesting, tick infestation ×migratory status of birds).

Line 156-157-This is part of result section, not materials and methods

3.2.Screening of arbovirus infection

Line 179 what does „tick infested bird” mean?

Line 179 Was this obtained by methods described in lines 112-128?

Line 179: I have not found description of RNA isolation from ticks, so how did you know that ticks were not positive for arboviruses RNA?

That part of results should be shortened and presented in table/bar/chart- in text lots of data can look a little messy and, in my opinion, it is hard to focus on every single number. Again, is there any statistical difference between tick-infested and non infested birds?

Line 246 please provide df

Line 247-249 I don't understand which given do authors understand as statistically significant: tick infestation or seropositivity of TRBV. What kind of relationship? higher, lower, tick presence = seroposotive bird ?? Please clarify

Line 247-249 please see above

Line 250 „the text continues here” I do not understand

Line 260 citation style

Discussion

This section is too long and should be shortened

Line 263 missing reference

Line 273 in comparison with our previous study -missing reference

Line 274 „prevalence 26% higher among blackcaps” in comparison to…..and you cannot say higher without statistic analysis.

I have many others suggestions/ comments to this section, but in my opinion the discussion must be rewritten

Line 366-377 and 379-390 repetition

Reviewer 3 Report

General comments

The present study reports on serosurvey on tick-borne encephalitis virus, and the tick-borne Kemerovo and Tribeč orbivirus infections in tick infested and non-infested birds. The results indicated an increase in TBEV and TRBV seroprevalence in birds over the past several years. And is the first study to investigate KEMV infection in Slovak bird population. This study highlights the importance targeted monitoring of birds and vectors of arbovirus introduction into new geographic areas.

Defects:

1.     The introduction of the research provides confusing background, and I suggest it is better to remove the introduction of Lipovník virus (LIPV) appropriately.

2.     Page 3, line 106. The sera were collected into new tubes and stored with the blood clots at -20 °C until examination. I’d like to know how long have the samples been stored before tested. In this study, the RNA of the three viruses was not detected in the blood clots. One possibility is that the blood samples really don't have viral RNA, another may be that the samples were stored at -20 °C for too much time, leading the disappear of virus. It depends on the hold time of the samples before tested.

3.     Page 4, line 172-177. Since the ticks have been collected from the birds, why not test the RNA of these three viruses in the ticks? Which will help to reveal the prevalence of the three viruses in Slovak.

4.     Page 9, line 379. The present work can not highlight the importance of research on zoonotic arboviruses. Targeted monitoring of tick-borne arbovirus in birds is more suitable for this study.

5.     Page 6, line 250. Please delete the sentence “The text continues here.”

Round 2

Reviewer 1 Report

The authors have done a great job of improving the article.

only, before being published, the authors could include in the discussion new studies, or complementary studies to continue the study of these. Just as a discussion for future analysis and work.